# Management of Aesthetic and Functional Deficits in Frontal Bone Trauma

**DOI:** 10.3390/medicina58121756

**Published:** 2022-11-30

**Authors:** Mihai Dumitru, Daniela Vrinceanu, Bogdan Banica, Romica Cergan, Iulian-Alexandru Taciuc, Felicia Manole, Matei Popa-Cherecheanu

**Affiliations:** 1ENT Department, Carol Davila University of Medicine and Pharmacy, 050472 Bucharest, Romania; 2OMF Surgery Department, Bucharest Emergency University Hospital, 050098 Bucharest, Romania; 3Anatomy Department, Carol Davila University of Medicine and Pharmacy, 020021 Bucharest, Romania; 4Department of Pathology, “Carol Davila” University of Medicine and Pharmacy, 050096 Bucharest, Romania; 5Department of ENT, Faculty of Medicine, University of Oradea, 410073 Oradea, Romania; 6Department of Cardiovascular Surgery, “Prof. Dr. Agrippa Ionescu” Emergency Clinical Hospital, 011356 Bucharest, Romania

**Keywords:** bone, complications, frontal, management, otorhinolaryngology, trauma

## Abstract

Frontal bone trauma has an increasing incidence and prevalence due to the wide-scale use of personal mobility devices such as motorcycles, electric bicycles, and scooters. Usually, the patients are involved in high-velocity accidents and the resulting lesions could be life-threatening. Moreover, there are immediate and long-term aesthetic and functional deficits resulting from such pathology. The immediate complications range from local infections in the frontal sinus to infections propagating inside the central nervous system, or the presence of cerebrospinal fluid leaks and vision impairment. We review current trends and available guidelines regarding the management of cases with frontal bone trauma. Treatment options taken into consideration are a conservative attitude towards minor lesions or aggressive surgical management of complex fractures involving the anterior and posterior frontal sinus walls. We illustrate and propose different approaches in the management of cases with long-term complications after frontal bone trauma. The team attending to these patients should unite otorhinolaryngologists, neurosurgeons, ophthalmologists, and maxillofacial surgeons. Take-home message: Only such complex interdisciplinary teams of trained specialists can provide a higher standard of care for complex trauma cases and limit the possible exposure to further legal actions or even malpractice.

## 1. Introduction

The recent use of Big Data algorithms for analyzing populations for the incidence and prevalence of head trauma cases revealed that the percentage of nasal bone fractures declined, the number of frontal bone fractures remained somewhat constant, whereas those of orbital fractures increased from 2011 to 2016 in a study published based on the Korean population [1]. Moreover in another 8 years retrospective cohort orbital trauma was confirmed in 23.6% of cases with concomitant soft tissue injuries and high risk of loss of vision or ocular motility [2]. This type of trauma presents a male predominance and a mean age of admission of 30 years [3].

Even from the 17th-century trauma to the frontal bone implied medical and legal aspects and still nowadays the distinction between medical/forensic autopsy and anatomical dissections for scientific research can be challenging [4].

The mechanism of trauma production ranges from accidents (automobiles, involuntary falls), human violence (with blunt or sharp objects), or self-inflicted [5].

The force of the impact is important in producing frontal sinus fractures with the involvement of both anterior and posterior sinus walls, like in the cases of animal kicks [6].

Gunshots, although rare, have a clear legal aspect, and unfortunately, when the trajectory affects the frontal sinus the outcome is almost fatal [7].

Unfortunately, there are even some cases of self-inflicted trauma to the head and neck region, and suicide attempts should not be excluded when analyzing a case with frontal bone trauma [8,9].

Frontal sinus fracture management depends on the functional deficit and life-threatening lesions and due to the potential aesthetic implications, the appropriate course of action is still controversial [10].

The arsenal of possible surgical approaches begins with primary closures, the use of osteosynthesis materials, or autologous tissues, sometimes the limit being only the imagination and innovative spirit of the surgical team [11,12].

The complications after such fractures range from local wound infection to nasal bleedings, short episodes of cerebrospinal fluid (CSF) leaks, orbital involvement, and even aesthetic deficits in the long term [13].

We have reviewed the current state-of-the-art practice in the management of aesthetic and functional deficits in frontal bone trauma. Following these guidelines will help clinicians understand the possible future medical and legal implications of these cases and assure proper informed consent on behalf of their patients. Moreover, these complex cases require input from various specialties and cooperation between different surgical departments.

The novelty of the present review resides in the endeavor of uniting the input from multiple specialties to align the guidelines followed by different departments. The objective is to provide a unified response and management of the head trauma cases and diminish the exposure to malpractice. This unified response starts from the very moment of admission, when unfortunately, cases tend to undergo multiple consults or are supervised by the first specialist present into the emergency department. In many cases involving the eyesight the first responder is the ophthalmologist, but further successful management of the patient and surgical planning is better performed by the ENT surgeon or the OMF surgeon [14].

## 2. Classification of Frontal Bone Fractures

Frontal bone (FB) fractures are found in about 12% of craniomaxillofacial trauma patients. This kind of trauma is a high kinetic energy trauma and subsequently has a high risk of cerebral damage [15].

Anterior wall fractures may be classified as anterior wall fractures with no displacement; anterior wall fractures with displacement but intact frontal sinus outflow tract (FSOT); or anterior wall fractures with displacement and FSOT injury [16].

Posterior wall fractures may be classified as posterior wall fracture without displacement and no cerebrospinal fluid (CSF) leak; posterior wall fractures without displacement and positive CSF leak; posterior wall fracture with displacement and no CSF leak; or posterior wall fracture with displacement and positive CSF leak [17].

## 3. Complications of Frontal Bone Fractures

Infection of the sinus, which causes sinusitis, may give rise to serious complications due to the proximity of the frontal sinus (FS) to the cranial cavity, orbit, and nasal cavity. Complications can develop into orbital cellulitis, epidural abscess, subdural abscess, and frontal lobe abscess. These complications may develop immediately or later after the traumatic episode [18].

Immediate complications in the fractures of the superior wall of the orbit with the involvement of the FS may occur within up to 6 months from the trauma—frontal sinusitis, meningitis, brain abscess, cavernous sinus thrombosis, cerebrospinal leak, diplopia, blindness, limitation of eye movement, neurosensory deficiency in the territory of the supraorbital nerves [19]. Late complications can occur more than 6 months after the trauma—formation of mucocele or mucopiocele, late frontal sinusitis, secondary brain abscess, and aesthetic defects.

## 4. Management of Frontal Bone Fractures (FBF)

The aims of FBF treatment are the restoration of facial appearance, the restoration of skull integrity and protection of the brain, and the prevention of late complications. The most important factor in the management of FBF is the involvement of the frontal sinus (FS). Despite the relative frequency of FS injuries, there is no consensus about their optimal management and numerous treatment algorithms were published in recent years [20].

A complicating factor is the involvement of the nasofrontal duct (FND). Its obstruction can lead to mucus retention and late infectious complications like frontal mucocele. So, the therapeutic options for frontal sinus fractures depend on the involvement of the anterior and posterior walls, and the functional integrity of the frontonasal duct [21].

Conservative treatment is indicated in cases where the fractures are strictly limited to the anterior wall of the frontal sinus and it presents a minimal displacement, without involving the frontonasal duct [22].

Reduction and osteosynthesis in displaced fractures are necessary but maintain FND permeability [23]. Displaced anterior wall fracture usually leads to a simple aesthetic deformity (Figure 1). Such a fracture needs surgery for correction. We performed surgery using the trans-eyebrow approach (Figure 2).

When we have a complex frontal orbital fracture, it becomes necessary a coronal approach for reduction and osteosynthesis [24]. This enables a wide exposure of the fracture site with the possibility of repositioning the bone fragments in an anatomic position and insertion of osteosynthesis devices (Figure 3 and Figure 4). In the case of small bone fragments, individual positioning is difficult and the optimal approach is ablation and reconstruction with titanium mesh (Figure 5 and Figure 6).

Exclusion of the frontal sinus by obliterating it in case the FND permeability is compromised but the posterior wall is intact. In this case, all the remaining mucous lining of the frontal sinus is completely cleared, and the frontonasal duct is obliterated with bone and periosteum. The sinus is obliterated with muscle, fat, or bone—without a consensus in the literature on the materials used for this purpose [25].

The fourth method of treatment is the cranialization of the frontal sinus, namely the disintegration of the posterior wall of the frontal sinus, obliteration of the frontonasal duct, complete closure of the sinus mucosa, and obliteration of the sinus with bone, fat, or muscle [26]. Posterior wall fracture usually results from high-impact injury and bears a risk of placing the intracranial content in direct communication with the nasal cavity. In posterior wall fracture, it is compulsory a mixed team formed by neurosurgeon, maxillofacial surgeon, and ENT surgeon [27].

Cranialization of the frontal sinus is the most radical method of FS fracture management. the procedure with complete removal of the posterior table. In effect, it increases the volume of the anterior cranial fossa at the expense of cranialized FS, and the brain can expand into this additional extradural dead space. Because intracranial space is entered, and there is a possibility of encountering dura and brain injury, cranialization should always be performed in cooperation with a neurosurgeon [28].

Autogenous bone graft for FS obliteration was first described. Cancellous bone grafts, most often harvested from the ilium, have been widely used as a filler material. Cancellous bone promotes re-ossification from both the periphery of the defect and centrally. Another advantage of cancellous bone over adipose or muscle tissue for obliteration is that it is easier to distinguish radiographically in the postoperative period between resorption, infection, and mucocele formation. Much more comfortable and safer is to harvest bone chips from the adjacent calvarium. It can be done using a bone scraper. In case the harvested amount of bone is not sufficient for filling a large sinus, it can be augmented by an admixture of bone substitutes such as a demineralized bone matrix [29].

## 5. Management of Mucoceles after Trauma to the Frontal Sinus

The occurrence of a frontal mucocele after trauma is a complication that may appear late after the moment of the trauma, even after a decade. There were some cases described after 35 years of trauma moment. Given this aspect, the patient should undergo a periodic evaluation for local tumefaction, double vision, eyelid ptosis, and local pain at palpation [30].

Frontal sinus mucoceles are cysts with the wall neighboring the periosteum lined with respiratory epithelium and containing fluid transforming into sterile puss. They are affecting patients over fifty and the cause is disruption of the drainage pathway. Up to one-third of the cases appear after trauma [31].

Clinically the patients present with eyelid ptosis and deformity of the superior eyebrow appearing for a long period without local inflammation. Sometimes the patients were initially admitted to neurosurgery departments with the reconstruction of the defects using metal plates (Figure 7).

Treatment targets complete removal of the cyst, and closure of the sinus cavity and frontal-nasal duct with fascia, temporal muscle, and autologous bone. Usually, a coronal approach is preferred with osteotomy superior to the eyebrow arch to preserve local bony landmarks [32].

One important aspect is the involvement of the orbit roof. This bony landmark requires fixation by titanium plates (Figure 8). Commonly the patient evolution after surgery is uneventful without further impairment of the eye movements (Figure 9).

## 6. Medico-Legal Aspects

For underage patients the informed consent of legal guardians is compulsory, and the parents must be aware of the fact that the trauma itself and the reconstructive procedures performed in the hospital are likely to interfere with the normal development of the viscerocranium with unforeseeable long-term possible complications [33]. Possible litigations between the patients and the healthcare system may arise from the quality of the osteosynthesis materials used during the reconstructive procedures after trauma, although current titanium mesh implants seem to have a high safety profile [34]. Moreover, the team caring for the trauma patient should be as thorough as possible in documenting possible hidden lesions or secondary pathology to prevent future medicolegal prosecution [35]. Ultimately achieving both functional and good aesthetic results are desirable, but the patient should be aware of the fact that the pathology was an emergency with a life-threatening outcome and that the aesthetic aspects may be improved through further procedures after complete recovery [36].

## 7. Discussions

The most recent approach to managing head trauma cases is the use computational simulation to improve the outcome of surgical bone reconstruction. Moreover these complex cases associate multiple bone trauma at the level of the limbs [37].

Obviously there are various manufacturers of ostheosynthesis equipment, but in all the images used to illustrate various bone fractures reconstructions we used Stryker, Berlin, Germany, titanium products for osteosynthesis in CMF surgery [38].

One modern approach is the use of engineered nanomaterials for closure of fractures. However, there is debate regarding the potential for adverse effects and inform product usage for individuals whose ocular health may be compromised by injury, disease, or surgical intervention [39].

One limitation of the present review is the scarce availability of data regarding the mortality due to frontal bone trauma. The mortality in such lesions should be divided in pre-hospital mortality and in-hospital mortality. The later type should be also analised taking into account conservative treatment or complex surgical procedures [40].

## 8. Conclusions

Treatment options in cases with frontal sinus fractures depend on the involvement of the anterior or posterior sinus walls and the functional integrity of the sinus communication drainage duct. One option is the complete closure of the sinus cavity if the normal drainage cannot be reinstated. Thus, we assure a clear barrier between the viscerocranium and neurocranium preventing potentially deadly meningitis. These cases require extended follow-up for long-term complications even decades after the trauma. In our experience maintaining the patency of the FND and of the NLD ensures a low risk for developing long term complications such as frontal sinus mucoceles or obstructive dacryocystitis. Following the example of skull base surgery training programs designed for ENT-surgeons and neurosurgeons teams performing endoscopic four hand surgery, there is a need for training programs designed for teams reuniting one specialist from each field of medical practice involved in head trauma cases: ENT surgeon, ophthalmologist, OMF surgeon, neurosurgeon at least.

## Figures and Tables

**Figure 1 medicina-58-01756-f001:**
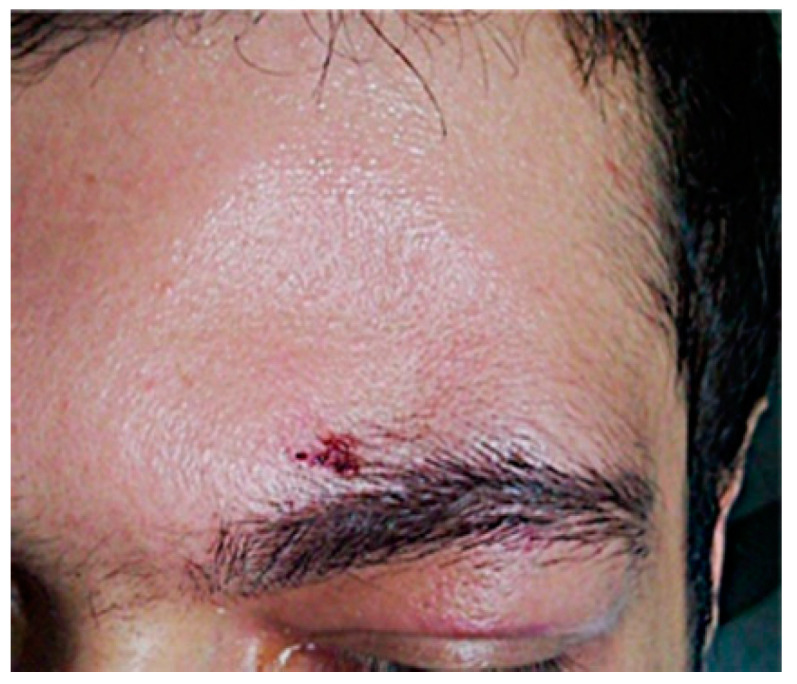
Left frontal sinus fracture with displacement.

**Figure 2 medicina-58-01756-f002:**
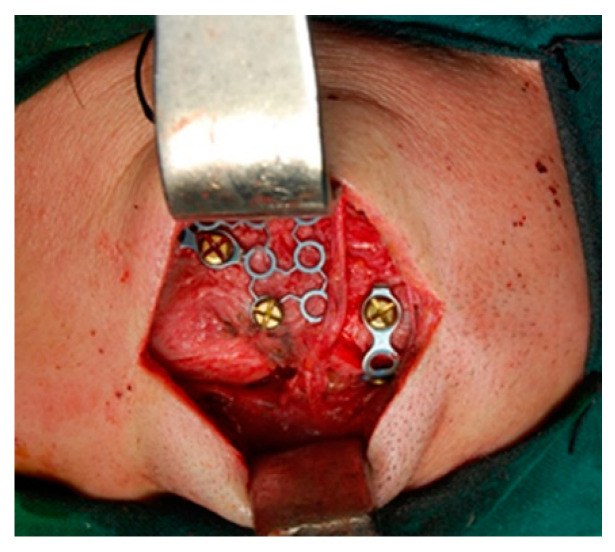
Osteosynthesis of a left frontal sinus fracture with displacement.

**Figure 3 medicina-58-01756-f003:**
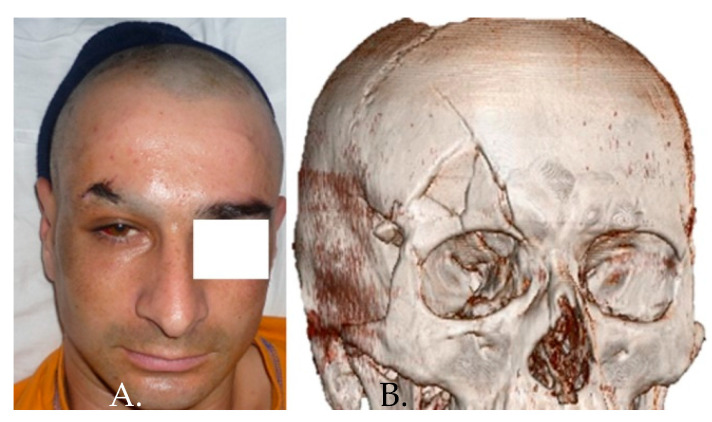
Human aggression using an axe resulting in right frontal sinus fracture with displacement—(**A**) clinical aspect; (**B**) 3D CT reconstruction.

**Figure 4 medicina-58-01756-f004:**
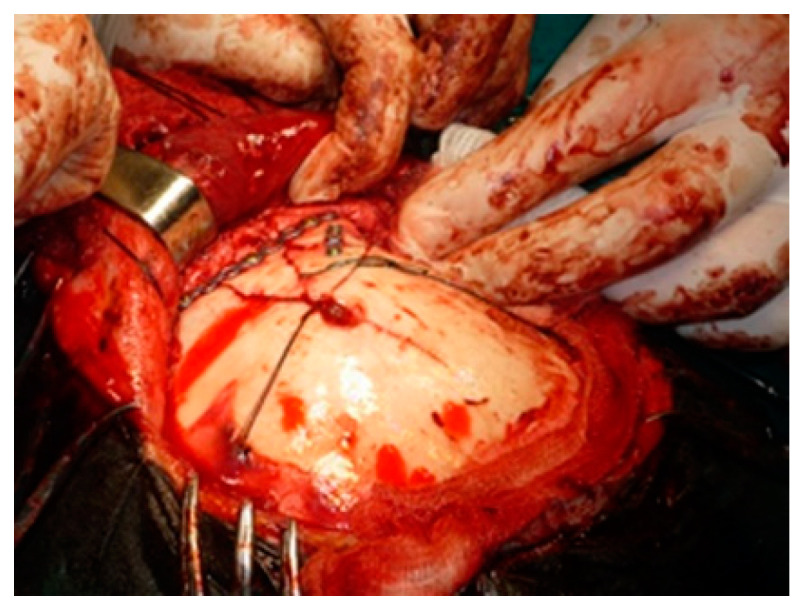
Coronal approach with surgical reduction and osteosynthesis of a right frontal sinus fracture with multiple fragments displacement.

**Figure 5 medicina-58-01756-f005:**
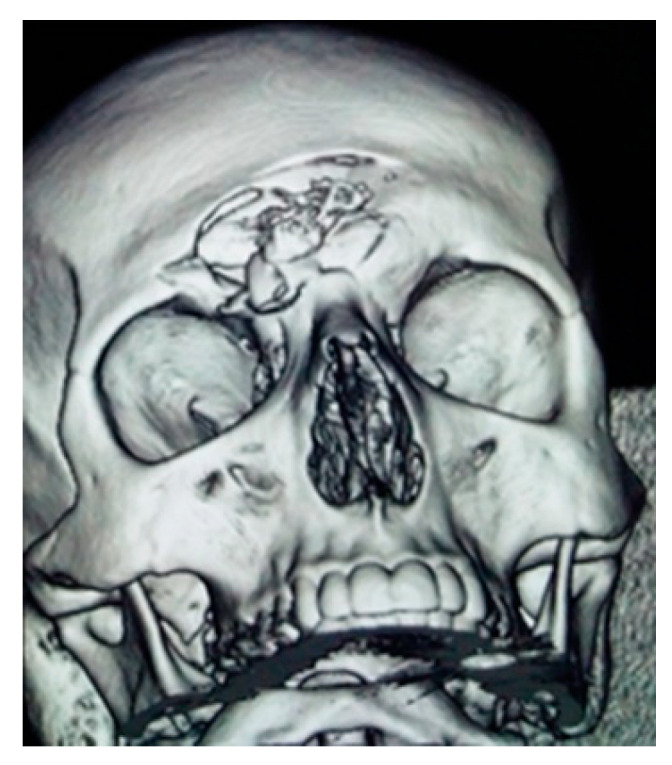
3D CT reconstruction of a right frontal fracture with displacement.

**Figure 6 medicina-58-01756-f006:**
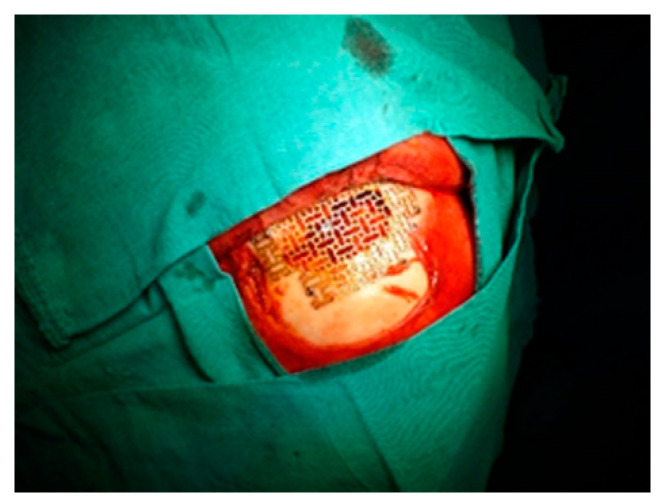
Surgical aspect of a right frontal sinus fracture with displacement benefiting from the use of a titanium mesh reconstruction.

**Figure 7 medicina-58-01756-f007:**
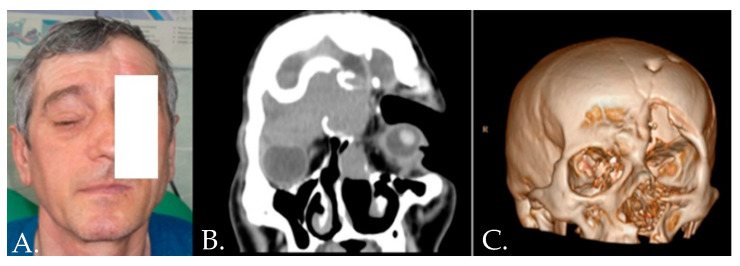
Right frontal sinus mucocele developed after trauma with right orbital involvement—(**A**) clinical aspect; (**B**) CT scan; (**C**) 3D reconstruction.

**Figure 8 medicina-58-01756-f008:**
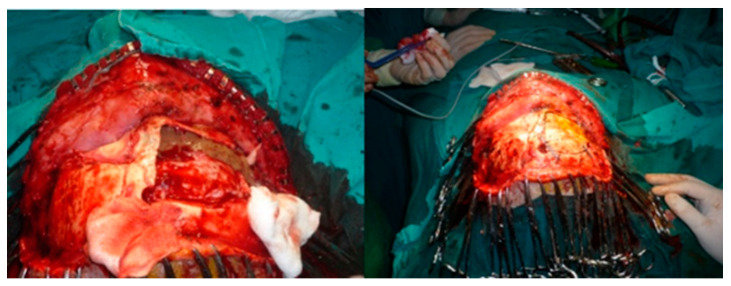
Giant right frontal sinus mucocele—surgical ablation, cranialization of the right frontal sinus, and reconstruction with bone flap.

**Figure 9 medicina-58-01756-f009:**
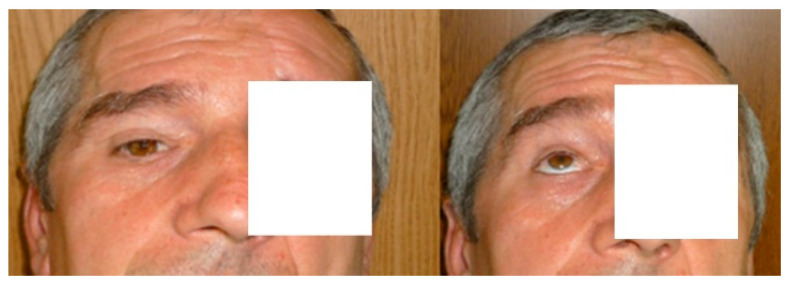
Recovery after surgical ablation of a post-traumatic mucocele, preservation of the esthetic aspect, and eye movements.

## Data Availability

All information presented in this review is documented by relevant references.

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
