# Peer review of "Management of Aesthetic and Functional Deficits in Frontal Bone Trauma"

_medicina, 2022, doi:10.3390/medicina58121756_

Round 1
Reviewer 1 Report
1. Please end your abstract with a "take-home" message.
2. Reorder keywords based on alphabetical order.
3. Novelty in the current review's is too weak. The past has seen an extensive literature of a lot of written material. It is required to provide more details for more explanation about the present novel in the introductory section.
4. Line 56, We review….., please make it into passive.
5. The authors encouraged to explain the role of computational simulation to impove human healthcare that would be recognize in bone trauma problem. The introduction and/or discussion part of an article should contain this crucial information. In addition, to support this explanation, the MDPI-suggested reference should be included as follows: Ammarullah, M. I.; Santoso, G.; Sugiharto, S.; Supriyono, T.; Wibowo, D. B.; Kurdi, O.; Tauviqirrahman, M.; Jamari, J. Minimizing Risk of Failure from Ceramic-on-Ceramic Total Hip Prosthesis by Selecting Ceramic Materials Based on Tresca Stress. Sustainability 2022, 14, 13413. https://doi.org/10.3390/su142013413
6. Manufacturer, country, and specification information for experimental setup should be presented with more specificity.
7. Error and tolerance of experimental tools used in this literature are important information that needs to be explained in the manuscript. It is would use as a valuable discussion due to different results in the further study by other researcher.
8. Please include the limitation of the present review, it is missing.
9. In the conclusion, please explain the further research.
10. Five years back literature should be enriched into the reference. MDPI reference is strongly recommended.
11. All through the manuscript, the authors created paragraphs that were only one or two phrases long, making the explanation difficult to understand. The authors should expand on their explanation to make it a more thorough paragraph. It is advised that one paragraph contain at least three sentences, with one sentence employed as the primary sentence and the other sentences employed as supporting sentences.
12. The manuscript needs to be proofread by the authors since it has grammatical and language issues.
13. After peer review, it is encouraged that a graphical abstract be included in the submission.
Reviewer 2 Report
PROS:
Comprehensive and well organised structure of the article with good and sugestive examples from your own experience.
CONS:
In the first paragraph you should assess the general prevalence of different types of skull fractures and then you should exemplify the prevalence in the Korean population according to your reference. This article is not about the Korean patients. Rephrase this paragraph.
The conclusions should reveal the reader what resulted from your own clinical experience with these cases (eg keeping the patency of the FND and of the NLD, not exposing the brain to the nasal cavity, etc) and give some exact and important landmarks for approaching these type of lesions
Round 2
Reviewer 1 Report
It is imrpoved, thanks/